# Influence of Extracellular Vesicles on Lung Stromal Cells during Breast Cancer Metastasis

**DOI:** 10.3390/ijms241411801

**Published:** 2023-07-22

**Authors:** Urvi Patel, David Susman, Alison L. Allan

**Affiliations:** 1Department of Anatomy & Cell Biology, Western University, London, ON N6A 5W9, Canada; upatel32@uwo.ca (U.P.); dsusman@uwo.ca (D.S.); 2Departments of Anatomy & Cell Biology and Oncology, Western University, London, ON N6A 5W9, Canada; 3London Regional Cancer Program, London Health Sciences Centre, London, ON N6A 5W9, Canada; 4Lawson Health Research Institute, London, ON N6A 5W9, Canada

**Keywords:** breast cancer, lung metastasis, pre-metastatic niche, extracellular vesicles, stromal cells

## Abstract

Breast cancer is a prominent cause of cancer diagnosis and death in women globally, with over 90% of deaths being attributed to complications that arise from metastasis. One of the common locations for breast cancer metastasis is the lung, which is associated with significant morbidity and mortality. Curative treatments for metastatic breast cancer patients are not available and the molecular mechanisms that underlie lung metastasis are not fully understood. In order to better treat these patients, identifying events that occur both prior to and during metastatic spread to the lung is essential. Several studies have demonstrated that breast cancer-derived extracellular vesicles secreted from the primary breast tumor play a key role in establishing the lung pre-metastatic niche to support colonization of metastatic tumor cells. In this review, we summarize recent work supporting the influence of extracellular vesicles on stromal components of the lung to construct the pre-metastatic niche and support metastasis. Furthermore, we discuss the potential clinical applications of utilizing extracellular vesicles for diagnosis and treatment. Together, this review highlights the dynamic nature of extracellular vesicles, their roles in breast cancer metastasis to the lung, and their value as potential biomarkers and therapeutics for cancer prevention.

## 1. Introduction

Recent global statistics from 2020 highlight that breast cancer is the most commonly diagnosed cancer and a leading cause of cancer mortality in women, with more than 2.2 million annual reported cases and more than 680,000 annual deaths worldwide [1]. Countries ranking high on the human development index are associated with higher incidence rates due to a combination of reproductive, hormonal, and lifestyle risk factors. There has been a rise in breast cancer cases in developing regions as they move towards improving human lifestyle and reflecting the profile of ‘westernized’ countries. In general, the 5-year net survival rates of breast cancer patients in high-income countries range from 85–90% for early stages (I and II), while lower income regions such as sub-Saharan African countries sit at ~66% [1]. These survival rates become lower for more advanced stages (III and IV) in both settings. Stage I and II tumors are primarily localized whereas stage III and IV breast cancers begin to form regional or distant metastases, respectively [2].

Breast cancers are often classified as invasive or non-invasive based on their histopathology, with ductal carcinoma in situ (DCIS) and lobular carcinoma in situ (LCIS) representing the two groups of non-invasive cancers. DCIS is subclassified into papillary, cribriform, solid and comedo groups, where papillary and cribriform subtypes are low-grade, and solid and comedo are high-grade. Treatment of DCIS is required due to the risk of progression into an invasive breast cancer [3]. LCIS is also considered a risk factor for potential disease progression and requires active surveillance alone when not associated with infiltrating carcinoma [3,4,5]. Breast cancer is further categorized into four main clinically relevant molecular subtypes based on expression of the estrogen receptor (ER), progesterone receptor (PR), and human epidermal growth factor receptor 2 (HER2). The subtypes, arranged by worsening prognosis, are luminal A (ER+, PR+/−, HER2-), luminal B (ER+, PR+/−, HER2+/−), HER2-positive (ER-, PR-, HER2+), and triple-negative (ER-, PR-, HER2-) [6]. Luminal A carcinomas are low-grade and responsive to hormone therapies such as tamoxifen or aromatase inhibitors with low rates of relapse. Luminal B cancers benefit from both hormone therapy and chemotherapy but are more aggressive and have higher recurrence rates. HER2-positive cancers can be subclassified as luminal HER2, expressing both hormone receptors and HER2; or HER2-enriched, expressing only the HER2 receptor. Therapies for this subtype include trastuzumab, pertuzumab, and other tyrosine kinase inhibitors which target the HER2/neu protein. Triple-negative breast cancers represent a heterogenous group of highly aggressive breast cancers. The lack of ER/PR/HER2 expression limits treatment options to individual or combined chemotherapy, surgery, and radiotherapy with reduced efficacy, although there are some promising new therapeutic options emerging including immunotherapy [7,8,9].

## 2. Breast Cancer Metastasis

Regardless of subtype, a significant challenge in treating breast cancer is that once the disease becomes metastatic, treatment efficacy substantially decreases [3]. As a result, over 90% of breast cancer deaths are attributed to complications that arise from metastasis [10]. Metastasis is a key hallmark of cancer that results in disease progression and the formation of secondary tumors in distant organs [10,11]. The metastatic cascade begins with cancer cells detaching from the primary tumor and invading into adjacent tissues; crossing the endothelial layer to intravasate into the bloodstream or lymphatic circulation [10,12]. These circulating tumor cells travel through the body to distant capillary beds and extravasate into the parenchyma of other organs [10,12]. At the secondary site, cancer cells proliferate under favorable conditions to generate new colonies and complete the metastatic cascade [10,12].

The distribution and growth of metastases in different organs is not arbitrary; instead, tumor cells demonstrate organ-specific preferences for metastasis [13,14,15,16]. In 1889, an English surgeon named Stephen Paget proposed the ‘seed and soil’ hypothesis, positing that a cancer cell (the ‘seed’) will only metastasize to secondary sites (the ‘soil’) that support and promote cancer growth. This proposed theory was inspired by Paget’s review of 735 necropsies, 241 of which possessed secondary liver metastases and 70 with lung infiltration. These findings highlighted that tumor dissemination was likely a non-random event, motivated by additional unknown factors at the time [14]. Paget’s hypothesis has largely withstood the test of time, generating more than a century of research that continues to elucidate the complex process of metastasis. Today we better understand that breast cancers demonstrate an increased propensity for metastasis to bone (50.7%), lung (23.9%), liver (19.7%), and brain (5.7%) tissue, with each breast cancer subtype demonstrating varying affinities for these targeted secondary organs [17]. Experimental studies from our lab using an ex vivo model system assessed the migration of breast cancer cells towards organ-conditioned media generated from the primary organ sites of breast cancer metastasis (bone, brain, liver, lung) of healthy mice. Breast cancer cells exhibited migratory patterns that parallel the organotropic metastatic behavior observed in breast cancer patients [18]. For example, patients with aggressive triple-negative disease typically show a greater occurrence of lung metastases [17]. Supporting this, we observed that triple-negative breast cancer cells demonstrated the greatest migration towards lung-conditioned media in vitro [18] and preferential spontaneous metastasis to the lung in vivo [19]. Further investigation revealed that the lung-conditioned media contained multiple soluble factors such as basic fibroblast growth factor (bFGF), epiregulin, and vascular endothelial growth factor A (VEGFA), which are involved in promoting tumorigenic and metastatic behaviors [18], supporting the idea that unique components of the lung microenvironment may support breast cancer metastasis.

## 3. The Pre-Metastatic Niche

Gao and colleagues [15] recently proposed five general principles underlying metastatic organotropism, all highlighting the importance of the metastatic microenvironment. These are (1) tropism may develop prior to dissemination by forming a supportive ‘pre-metastatic niche’ via modification of the microenvironment in the target organ; (2) chemotactic and adhesive factors aid in attracting cancer cells to and maintaining cancer cell survival and growth in a secondary microenvironment; (3) vasculature is essential to cancer infiltration via extravasation; (4) successful metastasis is dependent on the ability of resident stromal cells to modify the secretome and extracellular matrix (ECM) in the secondary microenvironment; and (5) cancer–microenvironment interactions perpetually change throughout the colonization process [15] (summarized in Figure 1). In particular, establishing a permissive microenvironment in the secondary site is a critical first step for successful metastasis. This concept of a ‘pre-metastatic niche’ (PMN) was first proposed in 2005 by Kaplan et al. [20], who noticed that the formation of aggregates of vascular endothelial growth factor receptor 1 (VEGFR1)-expressing, bone-marrow-derived hematopoietic progenitor cells in targeted organs of metastasis prior to cancer cell arrival at the site. Blocking VEGFR1 function limited pre-metastatic cluster formation and successful metastasis [20]. The PMN was subsequently defined as a microenvironment in a secondary organ that has been modified to improve cancer metastasis [21,22,23].

In the context of breast cancer, the lung is a particularly unique environment for metastasis given its physical proximity to the breast, extensive capillary networks and a large surface area for infiltration [15]. Additional modifications including remodeling of the lung ECM and secretome and breakdown of endothelial tight junctions all support formation of a lung PMN [24]. Current research is focused on better understanding the significance of the PMN in breast cancer progression. Growing evidence has established PMN formation as an integral step in organotropic metastasis of breast cancer. Generating the PMN demonstrates the successful recruitment and manipulation of multiple cell types and tissues prior to tumor dissemination. One of the key factors that can contribute to breast cancer–stroma crosstalk in order to mediate PMN formation are extracellular vesicles (EVs) [25,26].

## 4. Extracellular Vesicles

Extracellular vesicles are lipid-bound vesicles secreted by all cell types. Originally believed to be cellular debris, further analysis has revealed that EVs function as packaging entities containing molecular cargo such as proteins, nucleic acids, lipids, and metabolites, which can be transported through the body and transferred between cells as a mode of communication [27]. Extracellular vesicles are classified based on size, function, and biogenesis [28]. Exosomes (30–150 nm) are intraluminal vesicles that form from the plasma membrane via the endosomal pathway and have a regulated secretion [29,30]. Endosomes (100–500 nm) are degradative, pre-lysosomal vesicles that primarily function in recycling or degradative systems of ligands and macromolecules [31,32]. Early sorting endosomes (ESE) are initially formed by inward invagination of the plasma membrane, followed by maturation into late sorting endosomes (LSE), and a second invagination of the LSE to generate multivesicular bodies (MVB) [33]. Multivesicular bodies are a subset of endosomal vesicles that can be transported to the plasma membrane via cytoskeletal and microtubule networks. Multivesicular bodies are generated by inward invagination of the endosomal membrane with the release of discrete intraluminal vesicles (ILVs) into the MVB lumen. The MVBs can then either fuse with lysosomes and be degraded, or fuse with the plasma membrane and release the contained ILVs in the extracellular space as exosomes. Deriving from ILVs, exosomes have size constraints and cannot be larger than 150–200 nm, depending also on the method used to detect the size [28,33]. Microvesicles (30–1000 nm) are produced and released directly into the extracellular space via outward blebbing of the plasma membrane instigated by increased cytosolic calcium levels. Activation of calcium-dependent proteases such as calpain results in disruption of the cytoskeleton and subsequent vesicle release from domains called lipid rafts [29,34]. Microvesicles have selectively enriched contents and (similar to exosomes) have a regulated release, making them important signaling components in response to physiological changes [35,36]. Apoptotic bodies (50–5000 nm) are formed from dying cells as they degrade their cellular content [29,37]. They are formed via apoptotic cell disassembly, a general three-step process requiring plasma membrane blebbing, outward membrane protrusions, and fragmentation into EVs [38]. The mechanism of vesicle formation varies depending on regulators involved [39,40]. The release of apoptotic bodies serves as a mode of clearing dead cell debris via phagocytes and communicating with surrounding cells via the uptake of encapsulated proteins and nucleic acids. Finally, large oncosomes (1–10 μm) are exclusively cancer-related EVs. They are generated by the blebbing of amoeboid cancers and have been identified in luminal A and triple-negative breast cancer as well as in prostate cancer [41,42]. Unlike microvesicles and exosomes which have been the focus of more research, the full extent of apoptotic body and large oncosome signaling and functionality has not yet been elucidated [38,43].

A growing body of research highlights the importance of EV signaling in breast cancer progression, particularly in the context of the PMN and the metastatic microenvironment. This signaling is mediated in large part by the cargo contained within the EVs. As an example, in addition to proteins, EVs contain a variety of non-coding RNAs including circular RNAs (circRNAs), microRNAs (miRNAs) and long non-coding RNAs (lncRNAs). Along with other cargo components, non-coding RNAs contained within EVs contribute to many important aspects of the metastatic process, such as angiogenesis, cancer–stromal cell communication, and epithelial-to-mesenchymal transition (EMT) [44,45]. The protein and RNA cargo components that have been observed to be contained within breast-cancer-derived EVs are the focus of this review and are summarized in Table 1 and Table 2, respectively. Although lipids are important in regulating EV contents and EV-derived lipids can modify recipient cell activity [46], to date there is currently limited data in the literature focused on the role of lipid-based EV cargo in influencing breast cancer features. While work by Nishida-Aoki and colleagues has demonstrated that highly metastatic breast cancer EV lipid contents can activate angiogenesis upon uptake by endothelial cells [47], further research in this important area is needed.

Relative to the normal physiological state, the process of tumorigenesis has an impact on EV release, both in terms of the number of EVs released as well as their cargo/composition. In particular, the production and packaging of EVs is influenced by the conditions of the individual and various environmental stimuli. Cargo within EVs can be influenced by conditions such as obesity, cardiovascular disease, and autoimmune disorders to further support tumorigenesis. For example, EVs derived from obese adipose tissue promoted the proliferation of MDA-MB-231 and MCF7 breast cancer cells along with increasing invasion and migration of MDA-MB-231 cells. This was accomplished through activation of extracellular signal-regulated kinase (ERK) and phosphatidylinositol-3-kinase (PI3K) pathways [48]. There have been associations discovered between cardiovascular disease and high levels of microvesicles that have pro-angiogenic effects and influence inflammation [49]. Furthermore, autoimmune organs can produce EVs that carry autoantigens which trigger unwanted immune responses to assist tumor cells in successful immune evasion [50].

Extracellular vesicles are released at higher rates in environments of high temperature, acidity, and hypoxia, a feature that is particularly relevant in cancer since tumors are inherently hypoxic [51,52,53]. Once a tumor grows beyond a 70–200 μm proximity from nearby blood vessels, cancer cells become oxygen-starved [54]. Furthermore, research into the disparity between hypoxic versus normoxic cancer EVs has revealed that hypoxia affects EV size, production, and contents [55]. The altered signaling that is observed during hypoxia is largely driven by hypoxia-inducible factors (HIFs), a protein family of oxygen homeostasis regulators. These transcription factors exist in three variants (HIF1, HIF2 and HIF3) with an α or β subunit. The β subunit is constitutively expressed and minimally reactive to oxygen saturation, while the α subunit is increasingly expressed under hypoxic conditions. Oxygen deprivation promotes the heterodimerization of HIFα and HIFβ, resulting in transcription factor functionality and increased EV release [55,56]. With respect to breast cancer, HIF-1α has been associated with EV release. Pachane et al. [57] compared the proteomes of EVs derived from MDA-MB-231 breast cancer cells grown in hypoxic versus normoxic conditions and observed that hypoxic EVs were enriched with proteins associated with the mTOR, TGF-β and pro-angiogenic VEGFA/VEGFR2 pathways. In breast cancer, mTOR activation contributes to increased proliferation, migration, and invasion; TGF-β signaling is involved in the differentiation of resident fibroblasts into cancer-associated fibroblasts, and endothelial uptake of VEGF promotes angiogenesis [58,59,60,61]. Overall, these studies support the concept that hypoxia is an important physiological regulator of breast cancer EVs, resulting in EV-mediated cancer–stromal interactions and cancer-cancer signaling that support tumor growth and metastasis.

**Table 1 ijms-24-11801-t001:** Breast-cancer-derived extracellular vesicle protein contents.

Cellular Source of EVs	Experimental System	Protein	Function	Reference
MDA-MB-231 breast cancer cells (human)	In vitro	Tissue Factor	Exchanged between breast cancer cells to increase aggressiveness and induce cancer-associated thrombosis	[62,63]
	In vitro + in vivo	ITGα6, ITGβ1, ITGβ4	Promotes lung-tropic extracellular vesicles	[64]
	In vitro + in vivo	NDPK-B	Regulates purinergic signaling to enhance endothelial cell migration and permeability	[65]
	In vitro + in vivo	TβRII	Activates TGF-β signaling pathway to promote CD8+ T cell exhaustion and enhance EMT	[59]
	In vitro + in vivo	Survivin	Upregulates SOD1 to induce CAF activation	[66]
	In vitro + in vivo	Caveolin-1	Induces fibroblast-mediated tenascin-C release, M2-polarization of macrophages, and angiogenesis	[67]
	In vitro + in vivo	Myosin-9	Enhances macrophage infiltration	[68]
	In vitro + in vivo	MMP-1	Interacts with PAR1 to promote EMT, invasion, and migration of breast cancer cells	[69]
	In vitro	RRAGB, RPTOR, MTOR, RRAGA	Activates mTOR signaling cascades	[57]
	In vitro	SMAD2, SMAD3, SMAD9, SMAD1, SMAD5	Activates TGF-β signaling cascades	[57]
	In vitro	ABCF2, FXR2, AP2S1, SHC2, ARF6, ARF4, MTOR, CDC42BPB, STAM, SHC1, EIF3H	Activates VEGFA/VEGFR2 angiogenic signaling	[57]
	In vitro *+* in vivo	EphA2	Increases vascular permeability by downregulating tight junctions in endothelial cells	[70]
	In vitro *+* in vivo	EDIL3	Promotes breast cancer cell invasion via the integrin–FAK signaling pathway	[71]
MCF10CA1a breast cancer cells (human)	In vitro *+* in vivo	Annexin II	Promotes angiogenesis and activates p38, NF-κB and STAT3 pathways in endothelial cells	[72]
EO771 mammary carcinoma cells (mouse)	In vitro *+* in vivo	CCL2	Binds to CCR2-expressing cells in the lung and changes immune environment to increase metastatic burden	[73]

ITG, integrin; NDPK-B, nucleoside diphosphate kinase-B; TβRII, transforming growth factor-β type II receptor; TGF-β, transforming growth factor β; CD8, cluster of differentiation 8; EMT, epithelial to mesenchymal transition; SOD1, superoxide dismutase type 1; CAF, cancer-associated fibroblast; MMP, matrix metalloproteinase; PAR1, protease-activated receptor 1; RRAGB, Ras-related GTP-binding protein B; RPTOR, regulatory-associated protein of mTOR; mTOR, mammalian target of rapamycin; RRAGA, Ras-related GTP-binding protein A; SMAD3, mothers against decapentaplegic homolog 3; ABCF2, ATP-binding cassette subfamily F member 2; FXR2, FMR1 autosomal homolog 2; FXR2, fragile X-syndrome-related protein 2; AP2S1, adaptor related protein complex 2 subunit sigma 1; SHC2, Src homology-2-domain-containing-transforming protein C2; ARF, ADP-ribosylation factor; EIF3H, Eukaryotic translation initiation factor 3 subunit H; VEGFA, vascular endothelial growth factor A; VEGFR2, vascular endothelial growth factor receptor 2; EphA2, ephrin type-A receptor 2; EDIL3, EGF-like repeat and discoidin I-like domain-containing protein 3; FAK, focal adhesion kinase; NF-κB, nuclear factor kappa B; STAT3, signal transducer and activator of transcription 3; CCL2, C-C motif chemokine ligand 2; CCR2, C-C chemokine receptor 2.

**Table 2 ijms-24-11801-t002:** Breast-cancer-derived extracellular vesicle RNA contents.

Cellular Source of EVs	Experimental Model Type	RNA	Function	Reference
MDA-MB-231 breast cancer cells (human)	In vitro	miR-939	Targets VE cadherin to increase endothelial monolayer permeability	[74]
	In vitro + in vivo	miR-105	Targets ZO-1 to increase migration and permeability of endothelial cells	[75]
	In vitro + in vivo	miR-122	Suppresses glucose uptake in lung fibroblasts by downregulating the glycolytic enzyme pyruvate kinase	[76]
	In vitro + in vivo	miR-9-5p, miR-195-5p, miR-203a-3p	Targets ONECUT2 transcription factor to induce cancer stem cell phenotype and increase expression of genes associated with stemness in breast cancer cells	[77]
	In vitro + in vivo	miR-138-5p	Decreases KDM6B expression in macrophages, inhibits M1 polarization, and stimulates M2 polarization	[78]
	In vitro + in vivo	miR-9	Induces CAF phenotype	[79]
	In vitro + in vivo	circPSMA1	Inhibits miR-637, which targets Akt1 to regulate cell proliferation and migration in triple-negative breast cancer cells	[80]
MCF7 breast cancer cells (human)	In vitro	miR100, miR-222, miR-30a	Mediates drug resistance against docetaxel and adriamycin in sensitive breast cancer cells	[81]
	In vitro	miR-221/222	Targets estrogen receptor, mediates tamoxifen resistance in sensitive breast cancer cells	[82]
	In vitro	miR-155	Mediates chemoresistance against doxorubicin and paclitaxel, triggers EMT in sensitive breast cancer cells	[83]
MDA-MB-231, MCF7 breast cancer cells (human)	In vitro + in vivo	miR-146a	Modifies expression of thioredoxin-interacting protein and activates the Wnt/β catenin pathway, induces CAF phenotype	[84]
	In vitro	miR-1246	Targets CCNG2, promotes migration and viability of mammary epithelial cells	[85]
	In vitro	LncRNA-H19	Induces doxorubicin resistance in sensitive breast cancer cells	[86]
	In vitro + In vivo	LncRNA-SNHG1	Targets miR-216b-5p which upregulates JAK2 and STAT3 to enhance migration and angiogenesis of endothelial cells	[87]
4T1 mammary carcinoma cells (mouse)	In vitro + In vivo	miR-200b-3p	Binds to *PTEN* to regulate AKT/NF-κB/CCL2 cascade in alveolar epithelial type II cells and recruit myeloid-derived suppressor cells	[88]
	In vitro + In vivo	miR-183-5p	Targets PPP2CA to promote NF-κB signaling and enhanced expression of IL-1β, IL-6, and TNF-α in tumor-associated macrophages	[89]
4T07 mammary carcinoma cells (mouse)	In vitro + In vivo	Let-7	Recruit neutrophils and stimulate N2 polarization	[90]
4T1, 4T07 mammary carcinoma cells (mouse)	In vitro + in vivo	miR-125b	Negatively regulates p53, increases CAF activation markers	[91]
	In vitro + in vivo	miR-567	Increases sensitivity to trastuzumab and inhibits autophagy in resistant breast cancer cells	[92]

miRNA, Micro RNA; VE, vascular endothelial; ZO-1, zonula occludens 1; ONECUT2, one-cut homeobox 2; KDM6B, lysine demethylase 6B; CAF, cancer-associated fibroblast; circPSMA1, circular RNA proteasome 20S subunit alpha 1; EMT, epithelial to mesenchymal transition; CCNG2, cyclin G2; LncRNA, long non-coding RNA; JAK2, janus kinase 2; STAT3, signal transducer and activator of transcription 3; SNHG1, small nucleolar RNA host gene 1; PTEN, phosphatase and tensin homolog; NF-κB, nuclear factor kappa B; CCL2, C-C chemokine ligand 2; PPP2CA, protein phosphatase 2 catalytic subunit A; IL-1β, interleukin-1β; IL-6, interleukin-6; TNF-α, tumor necrosis factor α; Let-7, Lethal-7.

## 5. Influence of EVs on Lung Stromal Components

### 5.1. Endothelial Cells

One of the crucial stromal components in the lung is endothelial cells, which form a single layer lining along all blood vessels [93]. They are necessary for the process of angiogenesis—the formation of new blood vessels that supply nutrients and oxygen for tumor growth and also facilitate tumor cell dissemination [94]. Endothelial cells respond to pro- and anti-angiogenic factors released by the microenvironment that can either promote or inhibit vessel formation, respectively [93]. They also perform a barrier function, since substances in the blood are required to cross the endothelial layer in order to enter another site [93]. In the tumor microenvironment, the balance between these factors can be disrupted with an increase in pro-angiogenic factors to promote the formation of new vasculature [94]. Moreover, cancer-associated endothelial cells take on a different phenotype than that seen in normal tissues. In the normal physiological state, endothelial cells typically form organized and efficient vasculature with high integrity, allowing tightly regulated passage of nutrients to tissues and control of blood flow. In contrast, cancer-associated endothelial cells often form disorganized vasculature that is morphologically abnormal [95]. This vasculature tends to be unstable and leaky, influencing blood flow throughout the tumor [95] and supporting the metastatic process by easing the extravasation step in which tumor cells need to cross the endothelial monolayer to enter the secondary site [10].

There is growing evidence that tumor-derived EVs can promote angiogenesis by regulating the activity of endothelial cells at distant secondary sites to facilitate metastasis [54,96,97,98,99]. Zhou et al. demonstrated that EVs purified from triple-negative MDA-MB-231 breast cancer cells promoted metastasis-supporting behavior of primary human microvascular endothelial cells (HMVECs) [75]. They observed an increase in migration and permeability of the endothelial layer in vitro which was attributed to the presence of miR-105 in EVs derived from MDA-MB-231 cells [75]. The primary target of miR-105 is ZO-1, which was shown to be downregulated in HMVECs treated with EVs [75]. These results were replicated in vivo, where mice injected with MDA-MB-231 EVs showed similar pre-metastatic changes in lung endothelial cells, resulting in increased metastasis to the lung [75]. Another factor that regulates vessel permeability and contacts between endothelial cells are adherens junctions, specifically vascular endothelial cadherin (VE-cadherin) [74]. A study by Di Modica et al. showed that EVs from MDA-MB-231 cells were able to reduce the expression of VE-cadherin in endothelial cells, resulting in impaired endothelial function and enhanced permeability of the endothelial layer [74]. Endothelial cells treated with MDA-MB-231 EVs were found to display increased passage of breast cancer cells across the endothelium and a loss of cell-to-cell contacts in vitro [74]. This effect was attributed to the presence of miR-939 in these breast-cancer-derived EVs which targeted and disrupted expression of VE-cadherin [74].

In addition to miRNAs, growing evidence demonstrates that proteins contained within EVs can modify the behaviour of endothelial cells to promote angiogenic processes [72]. Specifically, breast-cancer-derived EVs have been shown to harbor annexin II (Anx-II), a protein associated with various cancer processes such as migration, proliferation, angiogenesis, and extracellular matrix degradation [72]. Maji and colleagues demonstrated that EVs from MCF10CA1a breast cancer cells contained Anx-II that resulted in increased endothelial tube formation in vitro and in vivo [72]. Moreover, when mice were injected with EVs from lung metastatic MDA-MB-4175 cells, the EVs localized primarily to the lung and resulted in higher numbers of lung metastases [72]. Another study comparing EVs from non-tumorigenic HME-1 cells to EVs from MDA-MB-231 cells demonstrated that MDA-MB-231 EVs were enriched in nucleoside diphosphate kinase (NDPK-B) expression and phosphotransferase activity [65]. Although NDPK-B is involved in several functions, its primary role is to transfer phosphate from nucleoside triphosphates to nucleoside disphosphates in nucleotide metabolism [100]. When endothelial cells were treated with NDPK-B+ EVs, enhanced migration and increased permeability of the endothelial monolayer was observed [65]. These changes were suggested to increase angiogenesis and extravasation in the lung to promote metastasis [65]. In mouse models, EV treatments showed an increase in pulmonary vascular leakage and greater lung metastasis that were caused by disruptions to the purinergic signaling pathway through changes in NDPK-B [65]. Together, these studies demonstrate the ability of breast-cancer-derived EVs to transport cargo to endothelial cells in the lung. By inducing pro-tumorigenic changes including increased permeability of the endothelial layer and enhanced angiogenesis, metastatic breast tumor cells are more likely to extravasate into the lung and colonize, leading to the successful formation of macrometastases.

### 5.2. Fibroblasts

Another important stromal component is fibroblasts, the primary source of ECM components in lung tissue. These cells generate optimal tissue conditions for lung function by synthesizing what is termed the “matrisome”. The matrisome is composed of all structural and adhesive proteins and ground substance components such as proteoglycans, glycoproteins, fibrillar proteins, and ECM-modifying proteins [101]. In addition, fibroblasts generate the basement membrane which separates the epithelium and surrounding stroma. This membrane serves as a barrier composed of type IV collagen and laminins with varying permeability, promoting the adhesion and migration of attaching cells via integrins and initiating cell signaling by releasing growth factors and other ECM-remodeling enzymes [102,103]. The ability of fibroblasts to create and remodel the lung ECM dictates interactions between all stromal cell components including endothelial cells, adipocytes, immune cells, neuronal cells, and others [104]. Additionally, the remodeling abilities of fibroblasts are essential for tissue repair during wound healing [105].

Thus far, the literature has emphasized the importance of fibroblast function in promoting cancer metastasis. The generation of heterogenous groups of cancer-associated fibroblasts (CAFs) has proven to be an essential step for remodeling the lung ECM into a cancer-promoting environment [106]. Research suggests that activation of the IL-6/STAT3, FGF2/FGR1, TGF-β/SMAD, and NF-κB signaling cascades may contribute to the activation and function of CAFs from normal fibroblasts [107]. Several studies have shown that during lung metastasis, there is significant crosstalk between primary breast tumor cells and lung fibroblasts that is mediated through EVs [108]. In breast cancer, specific RNAs and proteins contained within tumor-derived EVs such as miR-9, TGF-β, and Survivin activate CAFs [66,79,109,110]. Communication via EV release and uptake by fibroblasts supports the formation of a lung PMN and the establishment/growth of colonizing metastases.

Lung fibroblasts also support PMN formation by modulating the inflammasome and deposition of cancer-supportive ECM components. Hoshino and colleagues found that lung-targeted MDA-MB-231 EVs activated fibroblast S100 genes responsible for proliferation and migration [64]. In vivo studies have also shown that activation of S100A4 in mouse lung fibroblasts attracts T-lymphocytes which release cytokines that promote breast to lung metastasis. Deletion of S100A4+ fibroblasts have been shown to hinder metastasis, emphasizing the role of fibroblasts in inflammation and inflammatory cell recruitment [111]. Additional work by Gong et al. found that cyclooxygenase 2-expressing (COX-2^+^) resident lung fibroblasts promoted reprogramming of lung myeloid cells to reform the immune microenvironment. Deletion of the *PTGS2* gene encoding COX2 prevented fibroblast-mediated immune remodeling and subsequently impaired breast cancer metastasis to the lungs [112]. To further understand how breast cancers generate inflammatory, cancer-associated fibroblasts (iCAFs), Gonzalez-Callejo and colleagues treated lung fibroblasts with MDA-MB-231 EVs, then re-treated MDA-MB-231 cells with conditioned media acquired from EV-treated fibroblasts. They reported that EV treatment induced increased IL-6, IL-8, and CXCL1 secretion in fibroblasts, which corresponded with the acquisition of chemoresistance to paclitaxel and the expression of Nanog and ALDH1A1 stem cell markers in MDA-MB-231 cells [113]. These findings support the generation of a metastasis-promoting lung inflammasome organized by fibroblast recruitment of EVs.

A hallmark of PMN formation is remodeling of lung ECM components. Medeiros et al. [114] observed that mice bearing triple-negative SUM159 breast tumors demonstrated enhanced levels of periostin, fibronectin, tenascin-c, matrix metalloproteinase 9, collagen A1, C-C chemokine ligand 2 (CCL2), and lysyl oxidase in the lung when compared to the lungs of tumor-naïve or MCF7 tumor-bearing mice. Additional in vitro work has confirmed that lung fibroblasts only demonstrate elevated periostin and fibronectin levels when treated with triple-negative breast cancer EVs, and not luminal A EVs [114]. Earlier work by Libring et al. found that breast cancer EVs greatly improved the deposition of aberrant fibronectin clusters by lung fibroblasts consistent with fibronectin patterns at the primary tumor [115]. Increased ECM deposition at the primary tumor and PMN stiffens tissues to promote breast to lung metastasis [116,117,118]. Studies by Hoshino et al. suggest that lung fibroblast EV uptake is dependent on the presence of integrin β4 (ITGβ4), which is predominantly expressed in lung-targeting, triple-negative cancers [64]. Together, these findings suggest that lung ECM remodeling is linked to organotropic breast cancer EVs that prime the lung microenvironment for metastasis.

Aside from establishing the PMN, CAFs are important in the tumor microenvironment for cancer progression and can release their own EVs that contribute to this. CAF-derived EVs promote breast cancer growth and tumorigenicity [108]. Cancer-associated fibroblast EVs containing miR-500a-5p were transferred to MDA-MB-231 and MCF7 breast cancer cells. This resulted in the downregulation of ubiquitin-specific peptidase 28 (USP28) and induced breast cancer cell proliferation, migration, invasion, and EMT [119]. Patient-derived CAF EVs have been shown to contain miR-21, miR-143, and miR-378e and to promote EMT and stemness in T47D breast cancer cells [120]. Additionally, CAFs can reprogram breast cancer metabolism to promote an aggressive phenotype. CAF EVs containing lncRNA SNHG3 attenuated miR-330-5p activity to promote pyruvate kinase M1/M2 (PKM) function in support of enhanced glycolysis and proliferation of cancer in vitro and in vivo [121]. Interestingly, MDA-MB-231 EVs enriched with ITGβ4 activated mitophagy and lactate generation in CAFs, and CAF-conditioned media used to re-treat MDA-MB-231 cells enhanced breast cancer invasiveness, proliferation, and EMT [122]. These findings highlight the importance of breast cancer and CAF-derived EVs in the lung microenvironment to support critical aspects of successful metastasis such as proliferation, invasion, and EMT.

### 5.3. Immune Cells

At each step of the metastatic cascade, it is vital that tumor cells evade immune destruction to successfully survive and reach a distant organ for colonization [11]. Even at the new site, immune components are altered to form an immunosuppressive environment that facilitates the creation of the PMN. In the lung, two specific populations of macrophages exist including alveolar macrophages found in the airway and alveolar lumen, and interstitial macrophages found in the parenchyma [123]. These macrophages are part of the innate immune system and take on different phenotypes depending on their environment [124]. Typically, M1 macrophages are pro-inflammatory, have enhanced antigen presentation and exert anti-tumorigenic effects, while M2 macrophages are anti-inflammatory and promote tissue remodeling and tumor progression [124]. It has been observed that EVs from MDA-MB-231 cells carry miR-138-5p, which is transferred to macrophages to decrease the expression of lysine demethylase 6B (KDM6B). This ultimately leads to M2 polarization and transcriptional inhibition of pro-inflammatory factors involved in M1 polarization [78]. When mice bearing breast tumors were engrafted with macrophages treated with miR-138-5p breast-cancer-derived EVs, significantly higher incidences of lung metastasis were observed, suggesting that miR-138-5p may signal lung macrophages to induce pre-metastatic changes to the tumor immune microenvironment [78]. Another study revealed that MDA-MB-231 cells that exhibit high signal-induced proliferation-associated 1 (SIPA1) protein expression can release EVs with upregulated expression of myosin-9 to promote the recruitment of macrophages and metastasis of tumor cells to the lung [68]. Neutrophils are another component of the innate immune system that can undergo changes to promote tumorigenesis [125]. It has been demonstrated that Lin28B, an RNA binding protein, increased breast cancer stem cell populations which are a main source of EVs with low let-7 miRNAs [90]. These EVs were able to recruit neutrophils and encourage M2 conversion to build an immunosuppressive PMN in the lung mediated by programmed death-ligand 2 (PD-L2) upregulation and cytokine imbalance [90].

With regard to adaptive immunity, T cells play a crucial role in anti-tumor immunity; however, their activity can be stunted depending on the microenvironment [126]. EVs derived from human MDA-MB-231 and mouse 4T1 breast cancer cells have been shown to express programmed death-ligand 1 (PD-L1) on their surface, which inhibits T cell activation and cytotoxic functions to assist tumor immune evasion [127]. It has also been reported that MDA-MB-231 cells can release and transfer the TGF-β type II receptor (TβRII) through EVs to recipient cells to trigger the TGF-β signaling pathway [60]. The EVs carrying TβRII promote CD8+ T cell exhaustion by stimulating activation of SMAD3 (mothers against decapentaplegic homolog 3) to associate with T cell factor 1 (TCF1) transcription factor while also promoting EMT [60]. Nude mice injected with 4T07 breast cancer cells experienced significantly higher lung metastasis and lower metastasis-free survival when treated with TβRII+ EVs. This demonstrates the ability of EVs to establish an immunosuppressive PMN in the lung [60].

Cytokines are secreted or membrane-bound proteins that allow for intercellular communication [128]. The set of cytokines present in the tumor microenvironment have been shown to influence the behaviour of immune cells while promoting or inhibiting cancer pathogenesis [128]. Murine 4T1 breast cancer cells release LC3+ EVs which stimulate lung fibroblasts to produce CCL2 through the TLR2/MyD88/NF-κB (toll-like receptor 2/myeloid differentiation primary response 88/ nuclear factor kappa B) signaling pathway [129]. This in turn encourages lung PMN formation through recruitment of monocytes, suppression of T cell function, and enhanced vascular permeability. By reducing the release of LC3+ EVs from these cells or neutralizing CCL2, lung metastasis was inhibited, further supporting the role of EVs in developing the PMN [129]. Another group also found that CCL2 expression in the lung could be regulated by 4T1 breast cancer EVs containing miR-200b-3p, which binds to PTEN to stimulate the AKT/NF-κB pathway [88]. They observed EV uptake by type II alveolar epithelial cells in vivo which resulted in recruitment of myeloid-derived suppressor cells to promote immune suppression in the lung. The number of lung metastases was significantly higher in EV-treated mice compared to those with CCL2 knockdown, again demonstrating the role of EVs in priming the lung microenvironment for metastatic breast tumor cells [88]. Collectively, these findings demonstrate how EVs can modify the behaviour of different cells of the immune system to form the tumor immune microenvironment. These alterations come together to support PMN formation and assist metastatic breast tumor cells in successfully evading immune mechanisms in the lung.

Taken together, the studies described above highlight that EVs secreted by primary breast tumor cells can travel through the circulatory system to the lung in order to influence changes in the lung microenvironment via their effect on stromal cells such as endothelial cells, fibroblasts, macrophages, neutrophils, and T cells. In turn, these changes support formation of the PMN prior to the arrival of metastatic breast tumor cells to promote their survival and growth into successful metastases (Figure 2).

## 6. Clinical Implications

Current breast cancer treatments are non-curative for lung metastasis, and thus, it is imperative to explore diagnostics that may predict metastasis during early stages when it may be more treatable. Available imaging technologies may not be able to provide sufficient information regarding pre-metastatic changes occurring in the lung, and lung biopsies are invasive and may not be practical for the patient [130]. Liquid biopsies can be utilized as a way to measure EVs in bodily fluids such as blood that require less invasive procedures to retrieve, and can contain molecular factors that could act as biomarkers for PMN formation in the lung [131]. Additionally, in circumstances where tissue biopsies can’t be acquired, EV profiling can be performed to better understand the patients’ disease [132]. Molecular markers within EVs are protected within the lipid bilayer to maintain their stability and can include proteins, lipids, and nucleic acids that come from either cancer or stromal cells [29]. It has been shown that EVs collected from breast cancer patients have different molecular contents compared to healthy donors [132,133]. 

Several studies have analyzed blood serum levels of EV-associated miRNAs and their correlation to patient characteristics and prognosis. One study analyzed EVs of 50 breast cancer patients and 12 healthy women. They identifed higher levels of circulating EV-associated miR-373 related to enhanced survival in patients with aggressive triple negative breast cancer patients who are at a greater risk of lung metastasis compared to patients with luminal A/B disease [134]. By identifying miRNAs that are correlated with specific subtypes of breast cancer, better predictions can be made regarding potential metastatic destinations [134]. Zhou et al. extended their study by collecting circulating EVs in mice bearing MDA-MB-231 xenografts at a pre-metastatic or metastatic stage [75]. They observed significantly higher levels of EV-associated miR-105 in the sera of both groups of mice, indicating that the presence of miR-105 could be detected at early stages, prior to metastasis [75]. Additionally, they examined EVs derived from the sera of breast cancer patients and found that patients with significantly higher miR-105 levels eventually developed distant metastases [75]. Heat shock protein 70 (HSP70) has also been identified at increased levels within circulating EVs in metastatic patients compared to non-metastatic patients or healthy controls [135]. Overall, these studies demonstrate the potential of EVs as biomarkers for identifying patients at pre-metastatic stages [75,135]. Other markers such as developmental endothelial locus 1 (Del-1) and fibronectin have been identified on circulating EVs in the plasma of breast cancer patients [136,137]. These EVs were also observed to decrease substantially following treatment, suggesting a potential application for EVs in both identifying at-risk patients as well as following their response to treatment [136,137]. One clinical trial sought to identify the value of triple-negative, EV-derived miRNAs as biomarkers for disease recurrence. They discovered that patients with plasma containing EV miR-200a-3p, miR-203a-3p, and miR-7845-5p had an increased disease recurrence [138], and a few ongoing clinical trials are assessing the diagnostic and/or prognostic value of EVs in breast cancer (ClinicalTrials.gov: NCT05798338, NCT05831397, NCT05417048 and NCT04288141). Collectively, this work provides evidence that that EVs carry pertinent information that can be used to predict and monitor the state of breast cancer patients.

From a therapeutic perspective, there is growing number of studies exploring EVs as a mode of drug delivery for breast cancer patients due to their low immunogenicity, low toxicity, and high biocompatibility [139]. Since EVs secreted from metastatic breast cancers harbour integrins that selectively home to the lung, this can be exploited to deliver therapies directly to lung metastases [64]. For example, biomimetic nanoparticles coated with exosomal membranes from 4T1 mouse breast cancer cells have been constructed to hold siRNA-targeting S100A4, a protein that promotes metastasis and is involved in forming a favourable PMN for metastatic breast cancer cells in the lung. Initial in vitro studies showed efficient targeting of these nanoparticles to mouse embryonic lung fibroblasts and successful gene silencing of S100A4. Within postoperative lung metastasis mouse models, the accumulation of nanoparticles in the lung tissue and decreased numbers of pulmonary metastases was observed, demonstrating the potential of EV biomimetics as a therapeutic strategy [140]. Other studies have explored modifications to the bilayer of EVs such as engineering exosomes from human embryonic kidney cells (HEK293) to express the GE11 peptide or epidermal growth factor (EGF) to target breast cancer cells expressing the epidermal growth factor receptor (EGFR) [141]. Through this targeting strategy, exosomes have been shown to deliver let-7a miRNA to breast cancer cells in order to alter cell cycle progression and decrease cell division [141,142]. Intravenous injection of these exosomes in mice has demonstrated the potential to target both the primary breast tumor and metastases [141]. Another strategy involves loading EVs with chemotherapeutic drugs to better target tumor cells. Doxorubicin loaded into MDA-MB-231 exosomes through electroporation has been shown to inhibit breast tumor growth in mice by increasing its therapeutic index and efficiently releasing the drug to target cells [143]. Exosomal membranes derived from macrophages and further modified to target the hepatocyte growth factor receptor, which is highly expressed on triple-negative breast cancer cells, have been used to develop nanoparticles to also deliver doxorubicin to breast tumor tissues in vivo to reduce tumor growth [144,145]. The growth of pulmonary metastases has been shown to be inhibited by treatment with paclitaxel-loaded exosomes derived from murine RAW 264.7 macrophages [146]. Further advances within this field have led to novel strategies including bioengineering of tumor-derived exosomes with liposomes harboring lung-homing markers and gold nanorods. These have demonstrated improved therapeutic effects on lung metastases through a combination of thermal ablation mediated by gold nanorods to increase CD8+ T cells and cytokines in the lungs while also delivering paclitaxel to treat the tumor [147].

Currently, EV-based drug delivery remains in pre-clinical stages as more data need to be collected to assess safety and efficacy for patients [148]. Studies evaluating pharmacokinetics of EVs are increasing and are aimed at determining doses, route of administration, and targeting capacity. For example, mice that were intravenously injected with EVs from 4T1 cancer cells have shown quick clearance of EVs and low accumulation at the tumor site. However, intratumoral injections were found to be more efficient and EVs remained within the tumor tissue for a longer time [149]. The majority of EVs administered seem to accumulate in clearance organs such as the liver and spleen, which would influence the dose required to reach therapeutic doses at the target organ [150]. To date, no clinical trials involving therapeutic EVs in breast cancer have been published or are currently registered to ClinicalTrials.gov as ongoing. Nonetheless, with continued optimization and exploration of different uses for EVs in therapeutics, personalized treatment regimens for patients hold promise for improving clinical outcomes.

Although there have been promising studies supporting the use of EVs in the therapeutic setting, there remain several challenges that remain to be addressed, many of which parallel the challenges still faced in the experimental/pre-clinical setting [151]. For example, currently there is a large amount of heterogeneity in terms of EV isolation and purification protocols that are used, along with a lack of uniformity for evaluation standards regarding the quality of EVs [152,153]. There is also a lack of standardized procedures to store liquid biopsies for serial EV analysis, something that is very important given that storage time and temperature may influence the integrity of EVs, resulting in damage during freeze/thaw cycles [152]. Moreover, during EV isolation, both normal and tumor-derived EVs are found in the blood, which can cause interference when identifying biomarkers [154]. For drug delivery applications, a primary obstacle is the ability to mass produce EVs, as most isolation techniques are lengthy and have low yield. In addition, most methods of loading therapeutics into EVs have low efficiency and some can negatively influence the stability of EVs [139,155]. Finally, careful pharmacokinetic studies are needed to determine administration doses, route, and frequency for any new, EV-based therapies. Further work is needed to tackle these various challenges before EV-based biomarkers and therapies can be integrated into clinical use.

## 7. Conclusions

Breast cancer continues to be a global challenge and incidence rates are expected to rise in the next few decades. Metastasis is responsible for the majority of breast cancer deaths, as it disrupts the function of vital organs and currently lacks curative therapies. The lung is a deadly site for metastasis and remains a difficult site to treat. Although considerable advances have been accomplished in the field, there is an urgent need to understand the underlying molecular mechanisms that regulate breast cancer metastasis to the lung. Exploring the factors involved in forming the PMN and regulating the metastatic microenvironment is vital to this understanding. Growing studies have highlighted the role of EVs in promoting metastasis by acting as an avenue for communication between the primary breast tumor and the lung microenvironment. Changes to lung stromal components such as endothelial cells, fibroblasts, and immune cells have been shown to be mediated by cargo within EVs delivered to the lung. A primary limitation of the current literature in the field is the heavy reliance on specific breast cancer cell lines (i.e., MDA-MB-231) without validation in other models, including primary, patient-derived breast cancer cells. By expanding into other cell lines and more clinically relevant models such as patient-derived organoids, the heterogeneity of this disease can be better explored, and we can improve our knowledge regarding the role of EVs in lung metastasis. With this information, potential biomarkers can be identified for diagnosis, prognosis, prediction of therapy response, and tracking of disease progression. Due to the unique ability of breast-cancer-derived EVs to target the lung, novel strategies are being developed to use EVs as carriers of therapeutics. Altogether, elucidating the function of EVs will ultimately assist in formulating strategies to prevent or treat lung metastasis and improve breast cancer patient outcomes.

## Figures and Tables

**Figure 1 ijms-24-11801-f001:**
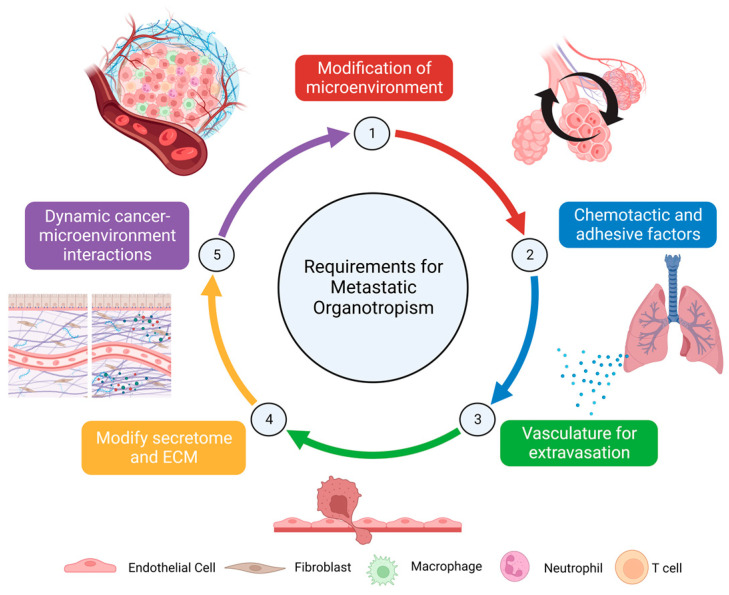
Requirements for metastatic organotropism and characteristics of the lung pre-metastatic niche. (1) Metastatic organotropism may develop prior to dissemination from the primary breast tumor to the lung by forming a supportive ‘pre-metastatic niche’ via modification of the lung microenvironment. (2) Chemotactic and adhesive factors aid in attracting breast cancer cells to and maintaining breast cancer cell survival and growth in the lung. (3) Vasculature within the lung is essential for cancer infiltration via extravasation. (4) Successful metastasis is dependent on the ability of resident stromal cells to modify the secretome and extracellular matrix (ECM) in the lung. (5) Breast cancer–lung microenvironment interactions perpetually change throughout the colonization process. Based on principles reported by Gao and colleagues [15]. Created using BioRender.

**Figure 2 ijms-24-11801-f002:**
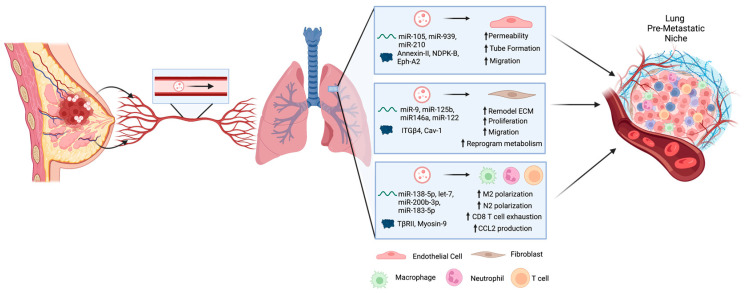
Extracellular vesicles (EVs) are secreted by primary breast tumor cells and travel through the circulatory system to the lung in order to influence changes in the lung microenvironment. These changes are mediated by cargo within the EVs including proteins and RNAs that can be taken up by various stromal cells including endothelial cells, fibroblasts, macrophages, neutrophils, and T cells. Altogether, these changes support formation of the pre-metastatic niche prior to the arrival of metastatic breast tumor cells to promote their survival and growth into successful metastases. Created using BioRender.

## Data Availability

Not applicable.

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
