# Peer review of "Influence of Extracellular Vesicles on Lung Stromal Cells during Breast Cancer Metastasis"

_ijms, 2023, doi:10.3390/ijms241411801_

Round 1

Reviewer 1 Report

Urvi Patel et al., describe the role of exosomes and microvesicles derived from breast cancer cells on metastasis, with emphasis on lung metastasis. The manuscript is informative, well written and easy to read. However, I have few comments:

1. In section 4, a small discussion should be added about the factors that can generate the secretion of EVs with greater pro-invasive and pro-metastatic capacity (example: immunological diseases, obesity, dyslipidemia, among others). 2. The authors describe the content of EVs at the mRNA and protein level and their association with metastasis. However, the information in the tables can be supplemented with EVs cargo molecules such as FAK and Src, kinases that modulate invasion and metastasis, both in vitro models (Ramirez-Ricardo, J. et al. “Role of Src/FAK in migration and invasion mediated by extracellular vesicles from MDA-MB-231 cells stimulated with linoleic acid.” Medical oncology, 2021) and breast cancer patients (Galindo-Hernandez, O. et al. “Elevated concentration of microvesicles isolated from peripheral blood in breast cancer patients.” Archives of medical research, 2013; Ramírez-Ricardo, J. et al. “Circulating extracellular vesicles from patients with breast cancer enhance migration and invasion via a Src‑dependent pathway in MDA‑MB‑231 breast cancer cells.” Molecular medicine reports, 2020).     3. Given the characteristics of the manuscript, I suggest adding an independent image detailing the characteristics of the lung PMN.

The manuscript is informative, well written and easy to read.

Reviewer 2 Report

In this review, Patel et al described influence of extracellular vesicles on lung stromal cells during cancer metastasis. This teview is quite interesting and provided extensive number of published studies.The manuscript is well written and comprehensive and English language and style are fine. 

Please consider the suggestions below:

- It would have been interesting to describe the communication mechanisms between cells using EVs.

- The paragraph on hypoxia is interesting but doesn't seem to have an obvious link with the rest. I propose to make it an independent paragraph, with more emphasis on the role of hypoxia in metastasis, particularly lung metastasis.

- - Could you elaborate in the manuscript on the use of tables 1 and 2, which are not described.

Reviewer 3 Report

The review article by Patel and coauthors addresses the influence of extracellular vesicles (EVs) on different stromal cells to induce a pre-metastatic niche in the lung and therefore support metastasis formation. Moreover, the authors discuss the clinical value of breast cancer-derived EVs as potential metastasis biomarkers as well as the use of EVs for drug-delivery.

The review article is well written and organized, however, I would suggest improving the description of some issues related to EVs, including EV biogenesis and classification, as well as EV clinical relevance.

Below are the specific comments:

Major

1)  I would suggest improving paragraph 4 on EV description, specifically:

-       Lines 141-142 and lines 146-147: I find the description of exosome biogenesis unclear, and I suggest modifying it.  I will try to recapitulate the different steps. First, the plasma membrane invaginates to form an early-sorting endosome (ESE) that can mature into a late-sorting endosome (LSE) and form a multivesicular body (MVB). MVBs are generated by inward invagination of the endosomal membrane with the release of discrete intraluminal vesicles (ILVs) into the MVB lumen. Then, the MVB can either fuse with lysosomes and be degraded or fuse with the plasma membrane and release the contained ILVs in the extracellular space as exosomes. Deriving from ILVs, exosomes have size constraints and cannot be larger than 150-200nm, depending also on the method used to detect the size.

-       Lines 147: Microvesicles can be as small as exosomes (30-50 nm) and arrive at 1000 nm, because, deriving from budding of the plasma membrane, they have not the same size restriction as exosomes.

-       Lines 151-153. Not only microvesicles but also exosomes have a regulated release

-       Line 153: Apoptotic bodies range in size from 50 to 5000 nm (Kakarla et al., 2020 (2020); doi.org/10.1038/s12276-019-0362-8

-       I would mention also oncosomes, among the type of EVs identified

-    The authors report the impact of hypoxia on EV release from breast cancer cells. I would highlight first of all that oncogenesis itself has an impact on EV release, increasing the number of EVs, both exosomes and microvesicles, and affecting their composition.

2)    Table 1 is very interesting and informative. I am wondering if it would be possible to add in the Table if the data refer to in vitro or in vivo experiments and, when not already specified, which is the cell type on which EVs exert the reported function. Moreover, since in the text the authors discuss the role of EV-associated miRNA before proteins, Table 1 could be also inverted accordingly.

3)  EVs contain also several bioactive lipids. Interest on EV-associated lipids is gaining increasing interest. I was wondering if, besides proteins and RNAs, EV-lipids have also been associated to breast cancer features. Maybe the authors could add to Table 1 also a section on lipids.

4)    It would be interesting if the authors could discuss the findings that triple negative breast cancer cells preferentially migrate toward lung.

5) Even though increasing evidence points to the diagnostic and therapeutic relevance of EVs, many issues have to be solved for EVs to enter clinics. Specifically, the authors could mention:

-       the challenges related to using EVs as biomarkers, such as the extreme heterogeneity of the isolation methods used, the lack of quality control, and the fact that tumor-derived EVs and normal EVs are mixed in the blood and this complicates identifying disease-specific biomarkers;

-       the lack of defined and standardized procedures to collect and store liquid biopsies for successive EV analysis;

-       besides the advantages of EVs as drug-delivery systems, due to low immunogenicity and targeting abilities, the challenges related to EV massive production, the finding of new EV sources, and low efficiency of EV loading and/or membrane functionalization;

-       pharmacokinetics studies to determine administration doses, route, and frequency.

6)    Lines 470-499: the authors should always specify which type of EVs has been used for drug-delivery

7)    The authors should mention the existence of any clinical trials related to the use of EVs in breast cancer as biomarkers or drug-delivery. If not present, please also mention this info in the paper.

8) In my opinion, it is a bit strange to see the only Figure in the Conclusions. I would suggest that the Figure be moved above.

9) I would suggest improving the bibliography, citing more articles on EV classification and description, EV role in cancer progression, including oncosome description (articles of of Meldolesi, Kalluri, Di Vizio), EV exploitation for biomarker analysis and drug-delivery, as well as the main position papers of the International Society of Extracellular Vesicles (ISEV), such as Thery et al., 2018.

Minor points:

-  line 258: Authors say “…a subsequent study by Modia et al…” but the reference is always the same, n.67, which corresponds to Di Modica and coauthors. Please check.  

-  please put genes in italics (i.e. line 327).

-  Line 391: signal-induced proliferation-associated 1 (SIPA1). Please invert. Check all the acronyms.

- Line 402: specify here that 4T1 cells are mouse breast tumor cells.

Only minor English editing required

Round 2

Reviewer 3 Report

The authors have carefully revised the manuscript according to given suggestions. I really appreciate their work.

I have only some minor comments:

Lines 183: please insert prostate cancers, since  the article of Minciacchi (ref. 44) refers to it.

Lines 196-197: please revise the sentence.

Tables 1 and 2 are no longer mentioned in the text.

Minor English editing.
